# MicroRNAs in Medullary Thyroid Carcinoma: A State of the Art Review of the Regulatory Mechanisms and Future Perspectives

**DOI:** 10.3390/cells10040955

**Published:** 2021-04-20

**Authors:** Francesca Galuppini, Simona Censi, Margherita Moro, Stefano Carraro, Marta Sbaraglia, Maurizio Iacobone, Matteo Fassan, Caterina Mian, Gianmaria Pennelli

**Affiliations:** 1Pathology Unit, Department of Medicine, University of Padua, Via Gabelli 61, 35121 Padua, Italy; francesca.galuppini@unipd.it (F.G.); marghe.moro91@gmail.com (M.M.); carrarostefano6@gmail.com (S.C.); marta.sbaraglia@aopd.veneto.it (M.S.); matteo.fassan@unipd.it (M.F.); 2Endocrinology Unit, Department of Medicine, University of Padua, Via Ospedale Civile 105, 35121 Padua, Italy; ssimonacensi@gmail.com (S.C.); caterina.mian@unipd.it (C.M.); 3Endocrine Surgery Unit, Department of Surgery, Oncology and Gastroenterology, University of Padua, Via Giustiniani 2, 35128 Padua, Italy; maurizio.iacobone@unipd.it; 4Istituto Oncologico del Veneto, IOV-IRCCS, 35128 Padova, Italy

**Keywords:** microRNA, medullary thyroid carcinoma, miR-21, miR-375, biomarkers, circulating microRNA

## Abstract

Medullary thyroid carcinoma (MTC) is a rare malignant neoplasia with a variable clinical course, with complete remission often difficult to achieve. Genetic alterations lead to fundamental changes not only in hereditary MTC but also in the sporadic form, with close correlations between mutational status and prognosis. In recent years, microRNAs (miRNAs) have become highly relevant as crucial players in MTC etiology. Current research has focused on their roles in disease carcinogenesis and development, but recent studies have expounded their potential as biomarkers and response predictors to novel biological drugs for advanced MTC. One such element which requires greater investigation is their mechanism of action and the molecular pathways involved in the regulation of gene expression. A more thorough understanding of these mechanisms will help realize the promising potential of miRNAs for MTC therapy and management.

## 1. Introduction

Medullary thyroid carcinoma (MTC) is a rare neoplasia derived from C cells secreting calcitonin (CT), and accounts for almost 5% of all thyroid malignancies [1]. Typically, MTC occurs sporadically in 75% of cases, and is hereditary in the remaining 25%, manifesting as hereditary–familial syndromes such as familial MTC (FMTC) and multiple endocrine neoplasia type 2A and 2B syndromes (MEN 2A and 2B) [2,3]. The molecular alterations underlying the hereditary forms are germline mutations in the rearranged during transfection (*RET)* gene [4]. This proto-oncogene is located on chromosome 10 and encodes a transmembrane kinase protein characterized by four extracellular cadherin-like cysteine-rich domains, an intracellular domain consisting of a juxtamembrane region, a tyrosine kinase domain and a carboxyl-terminus of variable length due to alternative splicing [5]. Point missense mutations at cysteine codons 611, 618, 620 and 634, which code for the cysteine-rich domain, trigger more than 90% of MEN-2A and 80% of FMTC cases [6]. Additionally, in more than 95% of MEN-2B cases, the main M918T mutation occurs in exon 16, which encodes the intracellular tyrosine kinase domain of the protein. This mutation induces increased ATP-binding affinity and the loss of autoinhibition, resulting in ligand-independent constitutive *RET* [7]. The different (intra and intercellular) point mutation sites are the main reasons behind the differential clinical and aggressive features of MEN-2A and MEN-2B syndromes. However, mutations affecting *RET* do not only distinguish hereditary MTC; more than 40–60% of sporadic MTC is associated with somatic *RET* mutations, in particular codons 883 and 918 [8]. Another gene involved in sporadic MTC is the *RAS* oncogene. Somatic mutations in *HRAS* and *KRAS* isoforms have been identified in 68% of sporadic MTC cases wild-type for *RET* mutations. Therefore, an alternative pathogenetic mechanism has been hypothesized, with *RAS* mutations often associated with favorable disease outcomes [9].

MTC is characterized by an indolent, slow-growing course, and patients with metastatic disease have a 5-year overall survival rate of approximately 40–50%. CT levels and TNM staging at diagnosis were previously the only definitive prognostic factors for MTC. Several investigations sought to determine how to separate MTC with a good prognosis from disease with metastatic progression. Prophylactic thyroidectomy is the standard treatment for inherited MTC, but for sporadic MTC, *RET* and *RAS* point mutations are often insufficient to distinguish tumors with worse prognoses.

MicroRNAs (miRNAs) are small single-stranded non-coding RNAs with an average 20–22 base pairs. They control gene expression at the post-transcriptional level and are among the most studied molecular prognostic factors [10,11]. Despite limited molecular information, the molecules are important in cell division, proliferation and longevity. The pathogenesis of different human cancers, including esophageal, breast, gastric, colorectal, pancreatic, lung and thyroid tumors, has been linked to the dysregulation of miRNA function, thus miRNAs act as tumor promoters or suppressors in cancer progression, by directly targeting genes involved in cancer cell biology.

## 2. Deregulated miRNAs in MTC

MiRNAs contribute to cellular homeostasis by controlling several post-transcriptional pathways. Studies have investigated miRNA signatures in differentiated follicular-derived thyroid cancer, with data showing extensive miRNA dysregulation [12].

In recent years, researchers have investigated miRNA actions in MTC development and progression. Nikiforova et al. identified a subset of ten upregulated miRNAs in two fine needle aspiration (FNA) MTC biopsy specimens [13]. Abraham et al. discovered a three-miRNA signature (miR-183, miR-375 and miR-9*) which distinguished between hereditary and sporadic MTC cases; miR-183 and miR-375 upregulation was identified as a molecular marker of extensive disease at diagnosis, as well as a poor prognosis biomarker [14]. Santarpia et al. reported a miRNA signature linked to metastatic MTC and biological processes such as epithelial mesenchymal transition (EMT) and the tumor necrosis factor-β-pathway [15]. Later, Mian et al. identified a subset of miRNAs (miR-127, miR-154, miR-224, miR-323, miR-370, miR-183, miR-375 and miR-9*) upregulated in MTC [16]. 

After these seminal studies, numerous molecular mechanisms involving miRNA dysregulation were not only identified in MTC processes, but were also correlated with disease prognosis, mutational status and responses to drug therapy in advanced disease stages.

In this review, we focus on the main miRNAs implicated in MTC and investigate underlying biological functions and prognostic implications. The main pathways involved in MTC and development, with regulatory functions of the most important miRNAs, are shown in Figure 1.

### 2.1. MiR-21

MiR-21 is one of the most important oncomiRs in carcinogenetic processes; its pro-oncogenic functions are gradually being unraveled, including identifying interactions with key RAS-ERK and RAS-PI3K signaling regulators, such as PTEN, RASA1, SPRED1, PDCD4 and SPRY1 [17]. By limiting translation of the GTPase activating protein (GAP), RASA1 and the NF1/GAP-associated factor, SPRED1, miR-21 maintains RAS-ERK signaling in breast cancer cells [18]. Interactions between endogenous miR-21 and miR-206 suppress GAP activities which are essential for maintaining wild-type RAS-GTP levels, RAS-ERK signaling and malignant properties, not only in cancer cells with wild-type RAS proteins, but also in tumor cells with RAS mutations. This GAP deficiency prevents RAS inactivation (i.e., the development of signaling-deficient WT-RAS-GDP), promoting RAS-ERK signaling, RAS-dependent cell phenotypes and carcinogenesis. MiR-21 functions in MTC have been well characterized; Pennelli et al. observed that MTC express significantly higher levels of miR-21 than normal thyroid cells [19]. One of the limits of the molecular analysis in MTC is the comparison between tumor tissue and non-tumor tissue due to the technical difficulties in obtaining normal C-cell samples to compare with MTC. These authors discovered a correlation between miR-21 upregulation and lymph node metastasis, advanced stages, and postoperative chronic disease, suggesting miR-21 acted as a pro-oncogenic miRNA during disease. Moreover, Pennelli et al. identified a negative association between miR-21 expression and immunohistochemical positivity for PDCD4, a tumor suppressor gene involved in apoptosis. They also demonstrated that there was no correlation between RET mutations and miR 21 expression while, interestingly, RAS mutations were associated with a greater expression of one of the miR-21 targets (PDCD4 and p-Akt pathway), indirectly confirming a correlation between RAS mutation and low miR-21 levels. Future studies will uncover underlying miR-21 intracellular mechanisms and determine therapeutic efficacy toward MTC.

### 2.2. MiR-375

Studies have investigated the relationship between miR-375 and breast, prostate, cervical, lung and esophageal cancer, but functional and regulatory mechanisms are limited. In MTC, miR-375 is a promising miRNA both as a prognostic factor and biomarker. Mian et al. [16] showed that MTC and C-cell hyperplasia were characterized by a massive upregulation in miR-375 levels, with an approximate 10.1-fold change with respect to non-tumor control tissue. Gundara et al. [20] reported that miR-375 expression levels were comparable between primary tumors and matched lymph node metastases. Moreover, in addition to overexpression in tumors, Galuppini et al. [21] reported that high miR-375 levels correlated with TNM staging, tumor progression and a worse patient outcome. 

Few studies illustrated the molecular mechanisms of miR-375 in MTC pathophysiology and progression. Hudson et al. [22] reported that levels of miR-375’s potential downstream targets, YAP1 and SLC16a2 were reduced in MTC, suggesting miR-375 could function as a negative regulator of YAP1 and SLC16a2. YAP1 deregulation was further confirmed by Galuppini et al. [21] by the immunohistochemical demonstration of YAP1 downregulation in MTC tissue, with reported high miR-375 levels. Lassalle et al. [23] discovered that SEC23A was a true miR-375 target. They also reported that improved sensitivity to vandetanib, a clinically important therapy for metastatic MTC, was related to increased miR-375 expression. These authors also indicated that miR-375 influenced cell proliferation and viability by increasing PARP cleavage and decreasing Akt phosphorylation. In terms of this latter process, Shi et al. [24] identified JAK2 and NGFR in the PI3K/Akt pathway as direct miR-375 targets in MTC, with downregulation observed at the mRNA rather than protein level. 

One reason why miR-375 is crucial to MTC management relates to its potential use as a biomarker in plasma. Romeo et al. [25] showed that higher plasma miR-375 levels were associated with distant metastases, and these levels were directly correlated with tumor burden. Furthermore, these authors observed reduced miR-375 levels in patients on vandetanib therapy, speculating miR-375 was a possible response biomarker to treatment.

### 2.3. MiR-224

MiR-224 functional studies in cancer are ongoing but not well-established. In lung adenocarcinoma, colorectal tumors, hepatocellular carcinoma and cervical carcinoma, the molecule appears to serve as a negative prognostic factor [26,27,28,29]. In contrast, MiR-224 overexpression is believed to be a predictor of increased radiosensitivity in glioblastoma and chemosensibility in prostatic carcinoma [30]. These data suggest miR-224 is important not only for cell proliferation but also for apoptosis, and the delicate balance between these processes determines the miR-224 phenotype in tumor cells. Cavedon et al. [31] demonstrated that miR-224 expression was significantly downregulated in MTC with high CT levels at diagnosis, advanced stage, persistent and progressive disease and disease-related death at the end of follow-up. Moreover, miR-224 was upregulated in sporadic MTC with somatic RAS mutations. These data reveal how elevated miR-224 levels are related to better prognoses and could represent an independent prognostic marker for MTC. 

The molecular mechanisms involved in this process have been fully elucidated. As previously reported, RET and RAS mutations are mutually exclusive during MTC. MTC-harboring RAS mutations showed a preferential activation of the PI3K/Akt/mTOR pathway, while RET-mutated MTC primarily involved the mitogen-activated protein kinase (MAPK) via [19]; thus, these different activation pathways may account for the different outcomes of the two molecular pathways. Importantly, in hepatocellular carcinoma [32], activation of the PI3K/Akt pathway by miR-224 was also demonstrated, suggesting this regulatory system is implicated in the biological effects of miR-224 in MTC. 

### 2.4. MiR-183

The upregulation of miR-183 was identified in sporadic MTC rather than in hereditary form; miR-183 over-expression was also associated with increased lateral cervical lymph node metastases rates, distant metastases and mortality [14]. Moreover, Abraham et al. reported that miR-183 expression values were higher in RET germinal mutations associated with worse outcome, but these differences were not statistically significant [14]. Similarly, miR-183 was related to oncogenesis in other cancers and downregulated the expression of the pro-apoptotic PDCD4 protein [33]. In 54 sporadic MTCs, miR183 was closely associated with lymph node metastasis [34]. MiR-183 together with miR-375 and miR-21 could help surgical choice when evaluating the requirement for lateral compartment neck dissections in the absence of clinical or radiological proof of nodal metastases, in addition to serum CT levels and disease staging.

### 2.5. MiR-127

MiR-127 has key roles in cancer cell proliferation, invasion and metastasis. The molecule was discovered as a tumor suppressor which targets BAG5 and controls cell proliferation, invasion and cell cycle progression in ovarian and pancreatic cancer [35,36]. Similarly, miR-127 inhibits osteosarcoma cell proliferation and migration [37]. Furthermore, by downregulating SEPT7, miR-127 acted as a tumor promoter in glioblastoma cell migration and invasion [38]. In gastric cancer, miR-127 blocked cell migration and invasion by upregulating Wnt7a [39]. In sporadic MTC, miR-127 expression was significantly lower in patients harboring somatic RET mutations than those with a wild-type RET status [40].

### 2.6. MiR-153-3p

Activation of the RET receptor tyrosine kinase is well-established as a key factor in MTC development and progression. MiR-153-3p is a RET-regulated miRNA which functions as a tumor suppressor, suppressing cell proliferation, invasion or migration, while promoting apoptosis [41]. MiR-153-3p was also implicated as a tumor suppressor in several other cancers [42,43,44]. Joo et al. reported that RPS6KB1 was a miR153-3p target via translational and post-translational repression, as well as repression of the downstream protein BAD. RPS6KB1 regulates critical biological processes, including cell development, proliferation, protein synthesis and cell cycle progression, as an effector of the mTOR signaling pathway [41]. The tyrosine-kinase inhibitors (TKI), cabozantinib and vandetanib, are promising novel therapeutics for advanced MTC. It was previously demonstrated that when inadequate inhibition of RET activity occurred after a TKI dose reduction due to side-effects, tumor inhibition by miR153-3p *via* mTOR signaling may provide a therapeutic advantage. 

### 2.7. The Long-Non-Coding-RNA—MALAT1

MALAT1 is a widely expressed long-non-coding RNA (lnc-RNA) with roles in alternative pre-mRNA splicing and epigenetic gene silencing [45]. MALAT1 is a marker of metastasis and poor prognosis in lung and other cancers. Chu et al. showed that MALAT1 expression was higher in MTC when compared with the normal thyroid, and that MALAT1 inhibition produced in vitro anti-oncogenic effects, including reduced tumor cell proliferation and invasion [46]. The pro-oncogenic activity of MALAT1 was enhanced by its ability to regulate the cell cycle-related transcription factors, B-MYB and p53 [47]. MALAT1 also stimulates EMT by epigenetically silencing E-cadherin expression [45,48].

### 2.8. MiR-31-3p

Several human cancers, including colorectal, oral and cervical cancer, have been linked to miR-31-3p dysregulation [49,50,51]. In MTC tissue and cell lines, Jiang et al. identified the major downregulation of miR-31-3p, and in both in vitro and in vivo assays, miR-31-3p overexpression greatly reduced MTC cell proliferation [52]. The molecular mechanisms underlying miR-31-3p actions involved RASA2 downregulation, which were confirmed by RASA2 knockdown where MTC cell proliferation was significantly inhibited. RASA2 serves as a GAP regulating RAS, which is one of the most highly mutated oncogenes in cancer, especially sporadic MTC [53].

### 2.9. MiR-34a and miR-144

MiR-34a, miR-34b and miR-34c are members of the miR-34 family which are primarily downregulated by DNA methylation in several cancers, and appear to function as tumor suppressors [54]. Several studies have demonstrated that miR-34a exerts direct actions by regulating the AXL protein in cancer cells [55]. The AXL receptor tyrosine kinase (AXL) is an upstream receptor of the PI3K/Akt/mTOR pathway and belongs to the TYRO3, AXL and MER class of receptor tyrosine kinases [56]. AXL was overexpressed in several tumors, with roles in cell survival, proliferation, migration, invasion and angiogenesis [57]. MiR-144 is another tumor suppressor miRNA with conflicting research findings. It was downregulated in certain tumors while upregulated in others [58]. Because of its suppressor nature, some studies have reported miR-144 downregulation and low expression levels [58,59]. Sun et al. [59] discovered that miR-144 downregulation in papillary thyroid carcinoma (PTC) was linked to E2F8 expression. Another study [60] observed that miR-144 expression was downregulated in human PTC tissue and cell lines, while Liu et al. [61] reported miR-144 expression was low in anaplastic thyroid carcinoma cells and tissue, and that cisplatin induced autophagy. MiR-144 is also upregulated in nasopharyngeal carcinoma tissue and cell lines [62]. In MTC tissue, Shabani et al. [63] reported the overexpression of miR-34a and miR-144 when compared with normal tissue, and moreover, they described AXL upregulation in clinical samples, while mTOR was downregulated. These elevated miR-144 and miR-34a expression levels suggest a possible role as biomarkers for MTC. However, in later work [64], although overall miR-144 and miR-34a expression was increased in MTC plasma samples, especially in RET-mutated cases, assay specificity and sensitivity was insufficient to recommend these molecules as new circulatory biomarkers for MTC. 

### 2.10. MiR-10a

From the first studies conducted on miRNAs in MTC, one of the top 10 most upregulated miRNAs in neoplastic tissues compared to non-tumor thyroid was miR-10a [13,22]. Santarpia et al. [15] reported miR-10a downregulation in MTC metastatic samples compared to primitive tumor samples. This article reported ten miRNAs that were significantly overexpressed and deregulated in metastatic tumors: miR-10a, miR-200b/-200c, miR-7 and miR-29c were downregulated while miR-130a, miR-138, miR-193a-3p, miR-373 and miR-498 were upregulated. MiR-10a, which is localized to chromosome 17q21 between HOXB4 and HOXB5 genes, regulates HOXD4 in breast cancer cells [65] and in acute and chronic myeloid leukemia [66,67]. Furthermore, miR-10a shares a high degree of sequence homology with miR-10b, which is a core regulator of breast cancer progression. In other carcinomas, miR-10a deregulation has been linked to metastases [68].

### 2.11. The miR-200 Family

MiR-200 family (miR-200b and miR-200c) member downregulation inhibits EMT by downregulating E-cadherin production in MTC cells, which endogenously expresses elevated miR-200 levels. By directly targeting ZEB1 and ZEB2, miR-200 downregulation increases tumor growth factor b (TGFb)-2 and increases tumor cell migration and invasion potential. These findings clearly indicate the miR-200 family has significant roles in controlling transition in the MTC cell phenotype, as cancer progresses [15]. MiR-200c expression was described as causative in the metastatic potential of MTC (21). Interestingly, it was observed [69] that the aberrant methylation of the putative promoter of miR-200c could underlie the differential expression in patients with distinct disease outcomes.

### 2.12. MiR-323

MiR-323 is a highly expressed miRNA in MTC [13], however functional studies are lacking. Recently, Ehyaei et al. [70] investigated miR-323 levels in correlation with RET mutational status. However, since no substantial variation in miR-323 expression was observed between the two types, this parameter cannot be used as a bio-index for germ line mutations in MTC. 

### 2.13. MiR-592

MiR-592 has been implicated in several human cancers [71,72]; e.g., high miR-592 expression was related to colorectal cancer tumorigenesis and poor prognosis [73,74]. MiR-592 inhibits the proliferation, migration and invasion of gastric cancer cells by repressing Forkhead box O3A [75], whereas miR-592 also promotes the proliferation, migration and invasion of colorectal cancer cells [76]. Thus, mi-R592 may be used as a carcinogenic biomarker for various cancers. Liu et al. [77] showed that miR-592 was upregulated in MTC tissue and cell lines when compared with normal samples. Moreover, CDK8 was identified as a miR-592 target gene, with miR-592 functioning as a negative regulator of CDK8. Several studies reported that CDK8 was a key oncogenic molecule in several human cancers, including colorectal, breast and prostate [78,79,80]. In MTC, miR-592 potentially functions as an oncogene by reducing CDK8 expression, making it a promising therapeutic target for MTC treatment.

### 2.14. MiR-9-3p

MiR-9-3p was significantly upregulated in MTC [16]. The molecule was also abnormally expressed and exhibited regulatory functions in several diseases, including trimethyltin-induced neurotoxicity, Huntington’s disease, glioma and primary brain tumors [81,82,83,84]. Chen et al. [85] observed that miR-9-3p upregulation exerted significant roles in MTC development by controlling cell growth and apoptosis by targeting BLCAP. Altered BLCAP expression inhibited cell survival and increased apoptosis in response to cytotoxic conditions. The BLCAP family regulates the p21(WAF1/CIP1) and Bcl-XL/Bcl-2 pathways, which are involved in caspase-dependent apoptosis. Bcl-2 protects cells from apoptosis by blocking the caspase 3-dependent proteolytic cascade and mitochondrial cytochrome C release [86,87]. 

### 2.15. MiR-129-5p

MiR-129-5p was discovered in a susceptible to mutation of chromosome 7q32 [88]. It acts as a tumor suppressor and is downregulated in gastric [89], colorectal [90] and hepatocellular carcinoma [91]. However, it is upregulated in laryngeal squamous cell carcinoma [92] and appears to act as an oncogene. Duan et al. [93] reported that miR-129-5p was downregulated in MTC tissue and cell lines. Using bioinformatics and luciferase reporter assays, RET was inhibited by miR-129-5p by directly binding to the 3′ untranslated region (UTR) of its mRNA. These findings suggested that higher miR-129-5p levels blocked the expression of RET and phosphorylated AKT, and that RET overexpression in cells with high miR-129-5p levels restored AKT phosphorylation status which had been decreased by miR-129-5p. Thus, miR-129-5p tumor suppressor activity was potentially linked to reduced *RET* expression and AKT phosphorylation suppression.

### 2.16. MiR-182

MiR-182 is a member of the miR-183 family (miR-183, miR-182 and miR-96) which are related to tumor growth in several malignancies, by mediating cell motility and migration [94,95,96]. MiR-182 overexpression exhibits highly aggressive characteristics, contributing significantly to increased invasion, survival and chemoresistance by repressing several targets, including FOXO3, MITF, MTSS1 and PDCD4 [97,98,99,100]. MiR-182 relevance in cancer is supported by Perilli et al., who proposed circulating miR-182 as a blood biomarker for tumor progression [101]. In comparison to normal thyroid tissue and cell lines expressing RET mutations, Spitschak et al. [102] discovered that miR-182 was majorly upregulated in tissue from a MTC cohort. HES1 was identified as a direct target of miR-182 owing to its binding to the 3′UTR, and was substantially downregulated in vitro and in MTC tissue. Biologically, the miR-182-HES1 axis controls thyroid cell migration and invasion, but not proliferation. Furthermore, HES1 loss was associated with lower Notch1 expression levels, suggesting a potential negative feedback loop mediated by *RET*-induced miR-182 expression.

### 2.17. MiR-222-3p and miR-17-5p

Zhang et al. [103] showed that miR-222-3p and miR-17-5p were significantly upregulated in plasma samples when compared with benign nodule patients and healthy controls. In thyroid cancer, the expression profiles of miR-222 -3p and miR-17 in tissue and cells were investigated and miR-222 resulted upregulated in both tissue and FNA samples [104]. This miRNA and miR-221 targeted p27kip1, p57, PTEN, TIMP3 and c-KIT and may be essential for thyroid cancer development [105]. MiR-17-5p is a member of the miR-17-92a cluster, and is upregulated in thyroid cancer and several other cancers, with key roles in a variety of pathological processes. This cluster, which contains seven miRNAs (miR-17-5p, miR-17-3p, miR-18a, miR-19a, miR-20a, miR-19b and miR-92-1) was overexpressed in anaplastic thyroid cancer cell lines [106].

The main miRNAs and their putative mechanisms in MTC are summarized (Table 1).

## 3. MiRNAs as Biomarkers in MTC and Future Perspectives

Circulating miRNAs are blood-based, and have been proposed as diagnostic, prognostic and therapeutic response biomarkers for disease [107,108]. In patients with intra-thyroidal disorders, the usual surgical procedure of complete thyroidectomy and neck dissection invariably results in remission. Consequently, many patients with persistent MTC require multiple therapies. The clinical history of these patients is highly variable, ranging from metastatic diffusion with accelerated progression to death, to indolent/slow-growing disease which remains stable for lengthy periods, or spontaneously develops even decades after the first treatment. Patients with locally advanced or metastatic MTC which has progressed and/or acquired complications are candidates for systemic therapy using vandetanib or cabozantinib TKIs. These orally administered molecules compete for the binding site of ATP on the catalytic site of the receptor, and consequently inhibit protein phosphorylation in transduction pathways. The overall response rate to TKIs is approximately 30%, with disease stabilization a major factor [109,110]. The concomitant measurement of CT and carcinoembryonic antigen serum levels is essential for patient follow-up; however, in TKI-treated patients, serum levels are highly variable and often do not reflect therapy responses. 

For these scenarios, circulating cell-free miRNAs may be increasingly useful biomarker resources. However, it is important not to underestimate the technical problems which potentially interfere with circulating miRNA evaluation. Proper normalization for correct miRNA quantification is a key issue in circulating miRNA research. Several techniques are widely used, including normalizing miRNA concentrations to serum volumes, adding external miRNAs, or using endogenous miRNAs or small RNAs. Since external controls cannot reflect serum sampling variability, using external miRNAs may not be a viable option.

An interesting aspect that has not yet been examined is the correlation between circulant miRNAs and the *RET* mutational status. Only the study by Shabani et al. [64] reported not only the overexpression of miR144 and miR34a in MTC plasma samples, but also the upregulation of this miRNAs in *RET*-mutated MTCs.

In this review, we explored the importance of miRNAs as key MTC biomarkers: miR-375, miR-144, miR-34a, miR-222-3p and miR-17-5p were all highly expressed not only in MTC tissue but also at significant levels in patient plasma, suggesting promising roles as circulating prognostic and diagnostic biomarkers [25,64,103].

## 4. Conclusions

MiRNAs are potential effective diagnostic and prognostic resources in oncology. In the past ten years, these molecules have been extensively investigated in MTC to identify expression patterns, action mechanisms and possible biomarkers with an emphasis on advanced stage identification. In MTC tumors, miR-375 and miR-21 are the most commonly upregulated miRNAs. However, an important discovery was identifying correlations between their levels of regulation and the mutational status that distinguishes MTC. The MTC mutational landscape has primarily focused on the *RET* proto-oncogene, which is mutated in approximately 40% of sporadic MTC cases, and almost all hereditary MTC cases. Other crucial molecular events, including *RAS* mutations, have been found in approximately 35% of sporadic MTCs. The main miRNAs involved in MTC carcinogenesis and progression are implicated in two signal transduction pathways, the MAP kinase and mTOR pathways. Many microRNAs are related to the *RET* mutational status. Some, including miR-224, miR127 and miR-129-5p, are downregulated in *RET* mutated MTCs due to their tumor suppressor role and/or their preferential activation of the RAS-related pathway. Others, including miR-183, miR-153-3p, miR-144 and miR-34a, are overexpressed in MTC with *RET* mutation. While miRNAs’ involvement in MTC etiology is recent, progress is ongoing to understand their regulatory mechanisms.

MiRNAs are excellent candidate biomarkers for diagnostic, prognostic and predictive patient stratification because they can be rapidly and reliably detected in tissue and body fluids. Furthermore, recent increased awareness of miRNA functions in cancer biology has made them appealing and exploitable tools for innovative therapeutic approaches.

## Figures and Tables

**Figure 1 cells-10-00955-f001:**
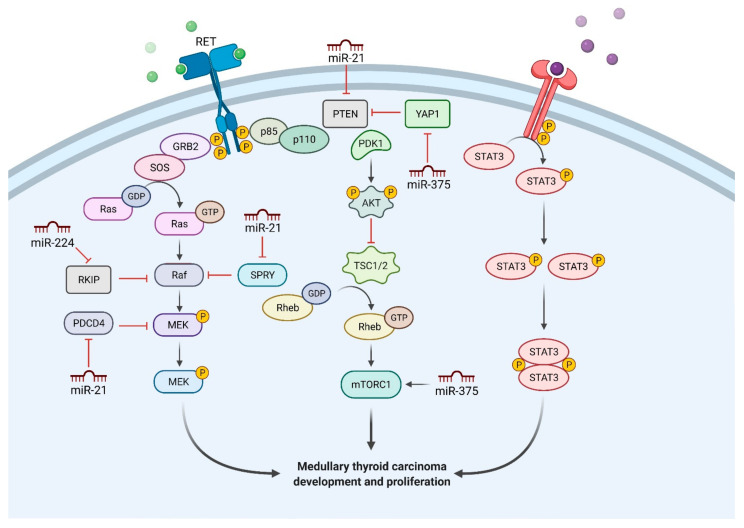
Molecular pathways involved in medullary thyroid carcinoma and development, showing regulatory functions of the most important miRNAs. Created using BioRender.com.

**Table 1 cells-10-00955-t001:** The main miRNAs involved in medullary thyroid carcinoma, including gene mechanisms and actions.

miRNA	Gene Mechanism	Actions
miR-21	Upregulates RAS/ERK pathway (targets RASA1 and SPRED1), inhibits PDCD4 and apoptosis	High levels associated with lymph node metastases, advanced stage and postoperative chronic disease; potential therapeutic target [19]
miR-375	Downregulates the PI3K/Akt pathway (targets YAP1, SLC16a2, SEC23A, PARP, JAK2 and NGFR)	Potential biomarker, plasma levels correlate with tumor burden, distant metastasis and response to vandetinib treatment [25]
miR-224	Dysregulates the PI3K/Akt pathway	Low levels in advanced MTC, high levels and positive prognostic factor in sporadic MTC with RAS mutations [31]
miR-183	Inhibits PDCD4 and apoptosis	High levels associated with lateral cervical lymph node metastases, distant metastases and mortality [14,34]
miR-127	Downregulates BAG5 and SEPT7, upregulates Wnt7a	Low levels in sporadic MTC harboring RET mutation [40]
miR-153-3p	Downregulates mTOR pathway (targets RPS6KB1)	Potential therapeutic effects in combination with tyrosine kinase inhibitors [41]
Long-non-coding-RNA—MALAT1	Downregulates B-MYB, p53, upregulates EMT (targets E-cadherin)	High levels in MTC, in vitro inhibition reduces tumor cell proliferation and invasion [46]
miR-31-3p	Downregulates RAS pathway (targets RASA2)	Low levels in MTC, reduces in vitro MTC cell proliferation [52]
miR-34a; miR-144	Dysregulates PI3K/Akt/mTOR pathway (targets AXL)	High levels in MTC, proposed as biomarkers but lack sufficient specificity and sensitivity [63,64]
miR-10a	Downregulates HOXD4	High levels in primary MTC but downregulated in metastases [15]
miR-200 family	Upregulates TGF-β2 (targets ZEB1 and ZEB2) and EMT (targets E-cadherin)	May correlate with metastatic potential [21]
miR-323	Unknown	High levels in MTC but not related to RET mutational status [70]
miR-592	Downregulates FOXO3a and CDK8	Potential therapeutic target [77]
miR-9-3p	Upregulates p21(WAF1/CIP1) and Bcl-XL/Bcl-2 pathway (targets BLCAP)	Enhances cell growth and inhibits apoptosis [16,85]
miR-129-5p	Down-regulates the PI3K/Akt pathway (targets RET)	Low level in MTC, acts as tumor suppressor [93]
miR-182	Downregulates FOXO3, MITF, MTSS1, PDCD4 and HES1/Notch1 pathway	Potential blood biomarker for tumor progression [101,102]
miR-222-3p; miR-17-5p	Downregulates p27kip1, p57, PTEN, TIMP3 and c-KIT	Potential blood biomarkers [103,104]

## Data Availability

No new data were created or analyzed in this study. Data sharing is not applicable to this article.

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
