# Peer review of "MicroRNAs in Medullary Thyroid Carcinoma: A State of the Art Review of the Regulatory Mechanisms and Future Perspectives"

_cells, 2021, doi:10.3390/cells10040955_

Round 1

Reviewer 1 Report

This is a very interesting and original review considering topic of high interest. There are some minor corrections needed:

  1. page 8 - title of the section 2.16. refers to MiR-129-5p, whereas it should refer to MiR-182
  2. page 9 - there is a misspelling in line 358; it should be "persistent" instead of "persitent"

Author Response

This is a very interesting and original review considering topic of high interest. There are some minor corrections needed:

  1. page 8 - title of the section 2.16. refers to MiR-129-5p, whereas it should refer to MiR-182.

Response 1: We thank the Reviewer for the Comment. We are really sorry for the inconvenience. We have modified the following subtitles in accordance with the suggestion: 2.14. MiR-9-3p (line 282 page 7) and 2.16 MIR-182 (line 305 page 7).

2. page 9 - there is a misspelling in line 358; it should be "persistent" instead of "persitent"

Response 2: In accordance with the suggestion, we have corrected the misspelling.

Reviewer 2 Report

The authors prepared a very comprehensive review on the role of miRNA in medullary thyroid carcinoma. Although the manuscript is well organized and the references are appropriate some improvements.

  1. In the introduction the authors state that “RAS mutations appear to be associated with an unfavorable outcome”. This is not true, several papers have been published demonstrating that RAS positive tumors are associated to a better outcome in particular when compared with RET positive cases (page 2, line 54). It is also known that RET positive cases have a bad outcome, thus the identification of this mutation at diagnosis can be a useful information for the follow-up of some patient. In addition the presence of e RET mutation is mandatory for the treatment of patients with target therapies
  2. Among others, authors report data on miR-21. According to these data, miR-21 are differentially expressed between tumor tissue and normal thyroid. This is a very “hot” issue. Normal thyroid is not the right control for MTC because MTC derives from a different type of cell. Are there any data on the differential expression of miRNA between RET positive and RET negative cases??
  3. The authors report data of Braham et al 2011 demonstrating that miR-183 is up regulated in MTC. Please indicate up regulated with respect to ???
  4. Are there any data on the correlation of the miR-183 and RET expression? And in general the other miR?
  5. The finding that miR-10a is down regulated in metastatic tissues with respect to the primary tumor is very interesting. Are there any other miR with a similar behaviour?
  6. The relevance of circulating miR as well as of circulating tumor DNA has been well demonstrated and few studies have been reported for MTC. Do the authors have any information on the correlation between circulating miR and RET mutations?

Author Response

The authors prepared a very comprehensive review on the role of miRNA in medullary thyroid carcinoma. Although the manuscript is well organized and the references are appropriate some improvements.

  1. In the introduction the authors state that “RAS mutations appear to be associated with an unfavorable outcome”. This is not true, several papers have been published demonstrating that RAS positive tumors are associated to a better outcome in particular when compared with RET positive cases (page 2, line 54). It is also known that RET positive cases have a bad outcome, thus the identification of this mutation at diagnosis can be a useful information for the follow-up of some patient. In addition the presence of e RET mutation is mandatory for the treatment of patients with target therapies

Response 1: We thank the Reviewer for giving us the opportunity to correct this error in the quoted sentence also based on the data we have shown. We have modified the sentence as follows: “Therefore, an alternative pathogenetic mechanism has been hypothesized, with RAS mutations often associated with unfavourable disease outcomes” (lines 52-54, page 2).

2. Among others, authors report data on miR-21. According to these data, miR-21 are differentially expressed between tumor tissue and normal thyroid. This is a very “hot” issue. Normal thyroid is not the right control for MTC because MTC derives from a different type of cell. Are there any data on the differential expression of miRNA between RET positive and RET negative cases??

Response 2: We thank the Reviewer for giving us the opportunity to deepen the topic. One of the limits of the molecular analysis in MTC is the is the comparison between tumor tissue and non-tumor tissue due to the technical difficulties in obtaining normal C-cell samples to compare with MTC. We add a phrase on this topic (lines 110-113 page 3).

There are few data on the correlation between miR-21 and the mutational profile of RET. In our previous work [19] we demonstrated that there was no correlation between RET mutations and miR 21 expression, while, interestingly, RAS mutations were associated with a greater expression of one of the miR-21 targets (PDCD4 and p-Akt pathway), indirectly confirming a correlation between RAS mutation and low miR-21 levels. We add a sentence about this topic (lines 117-121 pages 3-4).

3. The authors report data of Abraham et al 2011 demonstrating that miR-183 is up regulated in MTC. Please indicate up regulated with respect to ???

Response 3: We thank the Reviewer for the comment. Abraham et al reported an over expression of miR-183 in sporadic MTCs than in hereditary ones. We better specify this sentence. (lines 176-177 page 5)

4. Are there any data on the correlation of the miR-183 and RET expression? And in general the other miR?

Response 4: Abraham et al reported that miR-183 expression values were higher in RET germinal mutations associated with worse outcome, but these differences resulted not statistically significant. We add this information in the related chapter (lines 178-180 page 5).

As we have extensively reported in the text, many microRNAs are related to the mutational state of RET. Some including miR-224, miR127 and miR-129-5p are down-regulated in RET mutated MTCs due to their tumor suppressor role and / or their preferential activation of the RAS-related pathway. Others, including miR-183, miR-153-3p, miR-144 and miR-34a resulted over expressed in MTC with RET mutation. We add a paragraph in the Conclusion section about this topic (lines 384-388 page 10).

5. The finding that miR-10a is down regulated in metastatic tissues with respect to the primary tumor is very interesting. Are there any other miR with a similar behaviour?

Response 5: Santarpia et al (ref. 15) reported ten miRNAs that were significantly over expressed and deregulated in metastatic tumours: miR-10a, miR-200b/-200c, miR-7 and miR-29c resulted down-regulated while miR-130a, miR-138, miR-193a-3p, miR-373 and miR-498 were up-regulated. We add a sentences about this aspect in the Conclusion Discussion section (lines 257-260, page 6).

6. The relevance of circulating miR as well as of circulating tumor DNA has been well demonstrated and few studies have been reported for MTC. Do the authors have any information on the correlation between circulating miR and RET mutations?

Response 6: We thank the Review for the interesting comment. Shabani et al. (ref. 64) reported not only the over expression of miR144 and miR34a in MTC plasma samples, but also the upregulation of this miRNAs in RET – mutated MTCs. We add this comment also in chapter 3 (lines 372-375, page 10)

Reviewer 3 Report

  1. This manuscript is interesting and well-done.
  2. Theme of this review, to suggest a better understanding for mechanism of medullary thyroid cancer by micro RNAs. In this study, authors explain that the correlation between miRNA and medullary thyroid cancer and is well organized for readers to understand.
  3. Only minor points to be considered: English language should be revised throughout the text

Author Response

  1. This manuscript is interesting and well-done.

Response 1: We thank the Reviewer for the kind opinion.

2. Theme of this review, to suggest a better understanding for mechanism of medullary thyroid cancer by micro RNAs. In this study, authors explain that the correlation between miRNA and medullary thyroid cancer and is well organized for readers to understand.

Response 2: We thank again the Reviewer for the comment.

3. Only minor points to be considered: English language should be revised throughout the text

Response 3: We thank the Reviewer for the suggestion; English editing was reviewed by a native speaker.